# Transplanting Neural Progenitor Cells into a Chronic Dorsal Column Lesion Model

**DOI:** 10.3390/biomedicines10020350

**Published:** 2022-02-01

**Authors:** Kazuo Hayakawa, Ying Jin, Julien Bouyer, Theresa M. Connors, Takanobu Otsuka, Itzhak Fischer

**Affiliations:** 1Department of Neurobiology and Anatomy, Drexel University College of Medicine, Philadelphia, PA 19129, USA; julien.bouyer@drexelmed.edu (J.B.); Theresa.Connors@drexelmed.edu (T.M.C.); IFischer@drexelmed.edu (I.F.); 2Department of Orthopaedic Surgery, Graduate School of Medical Sciences, Nagoya City University, Nagoya 467-8601, Japan; t.otsuka@med.nagoya-cu.ac.jp

**Keywords:** neuronal progenitor cells, chronic spinal cord injury, cell transplantation, sensory system, axon regeneration

## Abstract

Cell transplantation therapy is a promising strategy for spinal cord injury (SCI) repair. Despite advancements in the development of therapeutic strategies in acute and subacute SCI, much less is known about effective strategies for chronic SCI. In previous studies we demonstrated that transplants of neural progenitor cells (NPC) created a permissive environment for axon regeneration and formed a neuronal relay across the injury following an acute dorsal column injury. Here we explored the efficacy of such a strategy in a chronic injury. We tested two preparations of NPCs derived from rat spinal cord at embryonic day 13.5: one prepared using stocks of cultured cells and the other of dissociated cells transplanted without culturing. Transplantation was delayed for 4-, 6- and 12-weeks post injury for a chronic injury model. We found that the dissociated NPC transplants survived and proliferated for at least 5 weeks post transplantation, in contrast to the poor survival of transplants prepared from cultured NPC stocks. The dissociated NPC transplants differentiated into neurons expressing excitatory markers, promoted axon regeneration into the injury/transplant site and extended axons from graft-derived neurons into the host. These results support the potential of NPC transplants to form neuronal relays across a chronic SCI, but they also underscore the challenges of achieving efficient cell survival in the environment of a chronic injury.

## 1. Introduction

Damaged axons in the mature mammalian central nervous system (CNS) show limited growth because of reduced intrinsic capacity for regeneration [1,2,3], absence of external factors and substrates that can support axon survival and extension [4,5,6], and the presence of inhibitory factors associated with inflammatory responses at the injury site [7,8]. Recent progress in cell and molecular biology identified mechanisms associated with the regenerative program in adult CNS neurons as well as the signaling associated with the inhibitory effects of Chondroitin Sulfate Proteoglycan and myelin paving the way for improved strategies to promote regeneration and repair after injury [9,10,11,12]. The chronic injury presents even greater challenges to regeneration because of additional complications emerging after the initial insult including the glial/fibrotic scar and inhibitory matrices in and around the injury site [13] and the varying degree of axonal retrograde degeneration and retraction [14,15]. Consequently, most of the experimental strategies that have been developed to promote axon regeneration have been directed to acute or subacute SCI, with less attention paid to the chronic stage despite the clinical reality that people with chronic SCI represent the vast majority of the patient population [16,17].

Cellular transplantation for SCI repair is a promising therapeutic strategy that utilizes a wide variety of cells, such as neural progenitor cells (NPCs) derived from embryonic or adult tissue or prepared by differentiation of embryonic or induced pluripotent stem cells (ES/iPS cells) from rodent or human tissue [18,19,20,21]. Transplantation of NPCs has been shown to provide beneficial effects in animal models of SCI by reducing the inflammatory responses and glial and fibrotic scaring, and by creating a permissive environment composed of neurotrophic molecules and growth promoting matrices [22,23,24]. In particular, NPCs are capable of replacing lost host cells that include neurons, astrocytes, and oligodendrocytes. These cells play a fundamental function in CNS homeostasis and replacement of these components is critical for reconstruction of the normal microenvironment of the spinal cord and for restoration of connectivity and function. We and others have previously demonstrated a neuronal relay can be established by NPC transplant across the injury site to reconnect the disrupted pathways in an acute SCI [25,26,27,28,29,30]. In such a relay formation, transplants of NPCs need to; (1) survive and differentiate into neurons, particularly excitatory neurons, (2) enhance host axon growth into the injury site, (3) support synapse formation between host-regenerating axons and transplant-derived neurons, and (4) extend axons out of the graft, reaching to a target [31]. Here, we studied the properties of NPC transplants in their potential application for a neuronal relay formation in a chronic injury model. We used a C4 dorsal column injury and transplanted NPCs prepared from rat spinal cord of embryonic day 13.5 (E13.5) at 4-, 6-, and 12-week post injury. The grafted cells showed good survival, differentiated into neurons, and promoted host sensory axon regeneration into the lesion/transplant site. Findings from this study establish the “proof of principle” for construction of neuronal relays with NPC as a cellular source for chronic SCI.

## 2. Materials and Methods

### 2.1. Animal Subjects and Experimental Design

Experimental design is shown in Figure 1A,B. Adult (225–250 g) female Sprague Dawley rats received a dorsal column lesion at C4 followed by a chronic transplantation of NPC or medium injection. All animals were given daily subcutaneous injections of cyclosporine A (10 mg/kg; Sandimmune; Novartis Pharmaceuticals, East Hanover, NJ, USA) that began 3 days before transplantation or medium injection. Buprenorphine (Webster Veterinary, Sterling, MA, USA) was used for pain relief every 12 h for 3 days and then as needed. Animals were divided into the following groups (Figure 1C): control: 4 weeks delay (*n* = 11), cNPCs (cultured NPCs): 4 weeks delay (*n* = 3); dNPCs (dissociated NPCs): 4 weeks delay (*n* = 10), 6 weeks delay (*n* = 5), 12 weeks delay (*n* = 5). All procedures were carried out in accordance with protocols approved by the Institutional Animal Care and Use Committee (IACUC) of Drexel University, following the NIH Guide for the care and use of laboratory animals.

### 2.2. Preparation of NPCs for Transplantation

We define two different cell preparations of NPCs, one as cultured NPCs (cNPCs) which were prepared from multiple embryos (E13.5 spinal cords, containing the stable alkaline phosphatase reporter gene (AP^+^) and pooled following short culturing and quality control for the composition of neuronal and glial precursors (NRPs and GRPs) at a ratio of 40:60 with >5% of either mature neural cells or non-neural cells [32]. They were then kept in vials of frozen stocks containing 3–5 million cells, and a second fresh stock of dissociated NPCs (dNPCs), was prepared as a cell suspension obtained from multiple pooled embryos (E13.5 AP + Fischer rat spinal cords) by mechanical dissociation of the spinal cords. As these cells are not cultured or processed, they are in fact fetal tissue and represent the same composition of the in vivo tissue. cNPCs preparation: following embryonic dissection, cells were plated on culture dishes coated with poly-l-lysine (PLL; 15 µg/mL; Sigma, St. Louis, MO, USA) and laminin (LN; 15 µg/mL; Invitrogen, Carlsbad, CA, USA) for 2–3 days in NRP (neuronal restricted precursor) basal media [DMEM/F12 (Invitrogen), bovine serum albumin (BSA; 25 mg/mL; Sigma), B27 (20 µg/mL; Invitrogen), N2 (10 µg/mL; Invitrogen), Pen-Strep (100 IU/mL; Invitrogen) supplemented with NT-3 (10 ng/mL; Peprotech, Rochy Hill, NJ, USA) and bFGF (20 ng/mL; Peprotech)] [26,33,34]. Cultured NPCs were collected in freezing medium (80% NRP basal medium, 10% chick embryo extract, 10% DMSO, 10 ng/mL bFGF and 20 ng/mL NT-3) and stored in liquid nitrogen. Two to three days before transplantation, cNPCs were thawed, cultured, and collected on transplantation day. dNPCs preparation: this method was based on a protocol used for acute preparation of dissociated segments of fetal spinal cord as previously described [35,36,37]. Briefly, the entire spinal cord from a single embryo was suspended in basal media and then mechanically dissociated sequentially with needles of different gauges (23, 25, and 27). Dissociated cell suspension was spun down and re-suspended in 4–6 µL of NRP basal media according to the size of isolated spinal cord. 2 µL of cell mixture (0.75−1.5 × 10^6^ cells per transplant) was injected into the lesion site immediately after the preparation without any additional factors.

### 2.3. Surgical Procedure

Rats were anesthetized by intraperitoneal injection of a cocktail of xylazine (10 mg/kg; Webster Veterinary, Sterling, MA, USA), acepromazine maleate (0.7 mg/kg; Webster Veterinary), and ketamine (95 mg/kg; Webster Veterinary). The hair over the cervical area was shaved, and the skin was cleaned and treated with iodine (Xenodine; Bipore Medical Devices, Northvale, NJ, USA). A laminectomy was performed at C4, and the dura was incised longitudinally above the dorsal columns to expose the spinal cord. A 30-gauge needle was used to make a complete unilateral injury in one side of the dorsal column by making a lesion cavity about 2 mm long, 0.5 mm depth, and 0.5 mm wide). The lesioned area remained empty with a 9–0 suture placed in the dura on top of the lesion center. This procedure facilitated locating the lesion center when delayed transplantation was attempted. For cell transplantation, at the appropriate time points, cNPCs and dNPCs were prepared and kept on the ice during the transplantation. The dura was incised at the sutured site, and 2 µL of cell suspension (1 × 10^6^ cells from cNPCs and 0.75–1.5 × 10^6^ cells from dNPCs per rat) was slowly injected into the lesion site using a 10 µL Hamilton syringe fitted with a 33-gauge needle. The dura was closed with a 9–0 suture. Animal care was performed as stated previously.

### 2.4. Cholera Toxin Subunit B (CTB) Labeling

To trace sensory axon regeneration, CTB was used to label the dorsal column axons. Three days before the end of experiment, animals were anesthetized with isoflurane at 4–5% (*v*/*v*) for induction and 2–3% (*v*/*v*) for maintenance, respectively. Sciatic nerves of both hindlimbs were exposed and CTB (2 μL, 1% in distilled water, List Biological Laboratories, Campbell, CA, USA) was injected into the sciatic nerve using a 10 µL Hamilton syringe with a 33-gauge needle. The incision was then closed. Animal care was performed as stated previously.

### 2.5. Tissue Collection

At different time points, animals were sacrificed with an overdose of euthasol (Webster Veterinary). Animals were perfused transcardially with 0.9% saline, followed by ice-cold 4% paraformaldehyde (PFA) in phosphate buffered saline (PBS). The spinal cords were removed and placed in 4% PFA overnight, and then transferred to 30% sucrose/0.1 M phosphate buffer at 4 °C for at least 3 days. The spinal cords containing the lesion and transplant site were removed, and embedded in M1 embedding matrix (Fischer Scientific, Pittsburgh, PA, USA). Embedded tissues were sagittally sectioned on a cryostat at 20 µm. Sections were collected on glass slides pre-coated with gelatin in series of six slides. Slides were stored at −20 °C.

### 2.6. Alkaline Phosphatase Histochemistry

Alkaline phosphatase (AP) histochemistry was used as previously described to visualize transplants and examine axon extension [38]. Sections were washed three times in PBS and then placed in PBS at 60 °C for 1 h to inactivate endogenous phosphatase activity. Sections were then washed in AP buffer (100 mM Tris, 50 mM MgCl_2_, 100 mM NaCl; pH 9.5), followed by staining in the dark for 2 h with AP staining solution [1.0 mg/mL nitroblue-terrazolium-chloride (Sigma), 5 mM Levamisole (Sigma), and 0.1 mg/mL 5-bromo-4-chlor-indolyl-phosphate (Sigma) in AP buffer]. All steps were at room temperature unless otherwise noted. Slides were coverslipped with FluoreSave^TM^ Reagent (Millipore, Billerica, MA, USA). Slides were viewed using a Leica DM5500B microscope (Leica Microsystem, Buffalo Grove, IL, USA) and images captured with an attached Hamamatsu ORCA ER digital camera using Slidebook imaging software 6.0 (Intelligent Imaging Innovations, Denver, CO, USA).

### 2.7. Immunohistochemistry

Slide mounted sections were washed three times in PBS (phosphate buffered saline) for 10 min and then blocked in 10% Goat or Donkey serum with 0.2% Triton X-100 in PBS for 1 h at room temperature. Sections were incubated with primary antibodies overnight at room temperature or 4 °C in PBS with 2% serum (See Table 1 for antibody sources and dilutions). On the next day, sections were washed three times in PBS and then incubated with secondary antibodies at room temperature for 2 h. Slides were washed three times in PBS, and coverslipped with anti-fade mounting media DAPI fluoromout-G (Southern Biotechnology, Birmingham, AL, USA). Slides were viewed with a Hamamatsu ORCA ER digital camera mounted on a Leica DM5500B fluorescent microscope with Slidebook imaging software or a Leica SP2 AOBS VIS/405 confocal microscope (Leica Microsystems Inc., Morrisville, NC, USA).

### 2.8. Analysis of the Longest Length of Extending Axons from Transplant

Sections with AP histochemical staining were used for this analysis. Images were taken at 10× magnification and montaged into single image to capture the entire range of AP^+^ axons. AP^+^ axons visualized in white matter were identified and the longest distance from the transplant center was measured in both the rostral and caudal directions. Processes were measured only when they were not associated with a cell body and when they were at least 100 µm in length so that only axonal, not glial processes were measured. All measurements were performed in an observer-blinded manner. AP transplant area was also measured from sections with AP histochemical staining by ImageJ.

### 2.9. Phenotypic Analysis of NPCs Transplant

To quantify differentiation of the transplanted dNPCs at 5 weeks after transplantation with 4 weeks delay, we used immunohistochemical staining of NeuN, GFAP, CC-1, and Nestin in combination with AP, and DAPI markers. Photographs were taken at 40× magnification. A minimum of 300 DAPI^+^ cells in the transplant site were counted from three different fields. Fields were randomly selected from white matter at the transplant site to eliminate over estimation of the NeuN^+^ ratio by contaminating host neurons in gray matter. The percentage of marker-expressing cells in total number of DAPI^+^ cells was calculated.

### 2.10. Scar Assessment

Lesions at the chronic stage of SCI form a distinct structure composed of glial and fibrotic scar tissue. To address the effects of NPC transplantation on scar formation, we evaluated both glial and fibrotic structures by immunoreactivity and volume size. *Glial scar*: Quantification of glial scar with GFAP^+^ staining was performed using ImageJ software as reported previously [39]. Sampling area was identified as a 100 µm^2^ area and the mean pixel value of GFAP^+^ staining from five sampling areas were averaged. Values were divided by background immune-labeled intensity, as averaged in five separate areas from intact tissue located 5 mm caudal to the injury site in the ventral white matter. Calculated values in each animal were obtained from three sections. Mean values for each animal were compared among groups. Light intensity and threshold values were maintained at constant levels for all analysis. *Fibrotic scar*: To assess the volume of fibrotic scar, immunohistochemical staining of PDGFRβ was performed [40]. Sections with PDGFRβ positive fibrotic scar was captured with 10× magnification. In all montaged images, fibrotic scar area on each section were measured and the scar volume was estimated using the Cavaileri principle defined as, Volume = ∑A × I_SF_ × *t* (A is the area, I_SF_ is the inverse of the sampling fraction, and *t* is the section thickness) [41]. Mean values for each animal were compared among groups.

### 2.11. Quantification of CTB-Labeled Axons

To determine the length of regenerating axons, sections were quadruple-stained with CTB, AP, GFAP, and DAPI and analyzed using NeuronJ software. An identifying lesion boundary was sometimes difficult to define due to modified glial scar by NPC transplants; therefore, we defined the lesion border using GFAP^+^ staining and DAPI^+^ staining, to indicate different cell components at the glial/fibrotic scar boundary. CTB^+^ axons were traced and then binned in a series of 200 µm boxes from the caudal edge of the lesion. Distribution of regenerating axons in transplanted groups was also referred to the caudal edge of the transplant site.

### 2.12. Statistical Analysis

All data are expressed as mean ± standard error of the mean (SEM). Statistical analysis was performed using SPSS statistical software (IBM, Armonk, NY, USA) using student *t*-test or one-way ANOVA followed by Bonferroni test for multiple comparisons, or SigmaPlot.13 software (SyStat Software, Inc., San Jose, CA, USA) using two-way ANOVA followed by repeated measurement with Holm-Sidak method for multiple comparisons. Significant difference was set at *p* < 0.05.

## 3. Results

### 3.1. NPCs Survive in Chronic Spinal Cord Injury

Our previous studies using cNPCs in acute dorsal column lesion without presence of additional factor showed good cell survival and integration at the lesion area with grafted cells filling the lesion cavity [26]. We therefore tested whether transplantation of cNPCs will also survive in chronic injury. We performed a dorsal column injury at C4 and transplanted cNPCs at 4 weeks after injury (1.0 × 10^6^/transplant). In control group, there was no AP^+^ staining around the lesion area (Figure 2A). In cNPCs group, AP histochemical staining showed that only few transplant-derived cells survived within the lesion site at 3 weeks after transplantation in some animals, while in others there was no cell survival at the lesion site (Figure 2B). In contrast, when dNPCs were transplanted 4 weeks after injury and survived for 3 weeks (Figure 2C), transplanted cells filled most lesion area and extended about 2 mm from the lesion core. Analysis of AP^+^ area in two groups indicates that more AP^+^ cells survived inside lesion area in dNPCs group than cNPCs group (*p* = 0.022, TTEST). Given the poor survival of cNPCs at chronic injury site, we then used dNPCs as transplants in all subsequent experiments.

Next, we tested whether the time of transplantation delay would affect the survival of transplanted cells because a long-term delay of the treatment can address patients at the late stage of the injury. We transplanted dNPCs at 4, 6, and 12 weeks after the injury and analyzed the animals at 5 weeks post-grafting. AP histochemical staining demonstrated that transplanted dNPCs survived well at lesion site, filled in the lesion cavity and extended into both rostral and caudal directions in all groups (Figure 3A–C). Transplants with a short delay of 4 weeks (4.17 ± 0.519 mm^2^) showed similar expansion compared to 6 weeks (3.45 ± 0.119 mm^2^) delay (*p* = 0.51), while transplants with 12 weeks (2.08 ± 0.531 mm^2^) delay covered a smaller area compared to 4 week delay (*p* = 0.033) but there was no significant difference compared to 6 week delay (*p* = 0.144). These data indicated that dNPCs transplant survived in chronic injury as late as 12 weeks after initial injury.

### 3.2. NPCs Differentiate into Mature Neurons and Extent Axons

To address cell differentiation of the transplants at the chronic injury site and in particular the neurons derived from the transplant, we performed immunohistochemical staining with lineage specific markers. Double staining with AP to identify transplanted cells demonstrated that cells inside the chronic injury site can differentiate into mature neurons (Figure 4A, AP/NeuN positive) as well as astrocytes (Figure 4B, AP/GFAP positive) and oligodendrocytes with AP/CC-1 positive staining were also found (Image not shown). Analysis of the percentage of differentiated cells inside the transplants (Figure 4C) at 5 weeks after transplantation showed that cell graft was composed of 19.80 ± 1.61% NeuN^+^ neurons, 47.57 ± 2.23% GFAP^+^ astrocytes and 23.50 ± 2.29% CC-1^+^ oligodendrocytes. There was only a small percentage of cells expressing nestin (7.39 ± 0.38%), a marker for immature progenitor cells. Importantly, transplant-derived neurons expressed vesicular glutamate transporters (VGLUT1 and 2) or glutamate decarboxylase 65/67 (GAD65/67), indicating the neurons have excitatory or inhibitory phenotypes (Figure 4D).

The immunohistochemical analysis also demonstrated axonal extension from the transplant site. Confocal images showed AP^+^ axons from transplants co-labeling with Tuj1, confirming the identity of the axons as being generated from transplant-derived neurons (Figure 4E). The length of the axons evaluated at 3 weeks post transplantation were over 2 mm in rostral and caudal direction from transplants (rostral 2.50 ± 0.76 mm, caudal 2.57 ± 0.50, *p* = 0.829, *t*-test, Figure 3C and Figure 4F). The length of AP^+^ axons at 5 weeks post transplantation increased (rostral 5.19 ± 0.28 mm *p* = 0.000, caudal 4.91 ± 0.49 mm *p* < 0.005, with one-way ANOVA with Bonferroni test, Figure 4F). Axon extension was observed in all groups, however, it was most prominent in the 4 and 6 weeks which showed similar extension to about 5 mm, while the 12 weeks delay showed less extension of about 3–4 mm (Figure 4F). These results might correlate to the smaller size of the transplant at 12 weeks delay as well as the increased inhibitory environment. Taken together, dNPCs transplanted in a 4–12-week delay differentiated into mature neurons with both excitatory and inhibitory phenotypes and extended axons, providing the basic elements for forming a neuronal relay in chronic injury.

### 3.3. NPC Graft Modification of the Glial/Fibrotic Scar

NPC transplantation into acute/subacute injury has been reported to attenuate reactive glial scar formation and reduce lesion size [42]. To address the effects of dNPC transplants on a glial and fibrotic scar which has already become established in the chronic phase after the injury, we evaluated GFAP immunostaining in the lesion interface and PDGFRβ^+^ area within the lesion site, respectively. In controls, strong GFAP^+^ cells (Figure 5A) were accumulated around the lesion area at 9 weeks post injury (4 weeks transplant delay). The lesion cavity was mostly filled by PDGFRβ^+^ cells (Figure 5D), indicating typical glial/fibrotic scar formation at the lesion area. In contrast, transplants significantly reduced GFAP+ staining (Figure 5B, with insert showed AP^+^ transplant) and PDGFRβ^+^ (Figure 5E) around lesion/transplant region at 5 weeks post transplantation (9 weeks post injury, 4 weeks transplant delay). Quantitative analysis revealed that GFAP staining intensity around the lesion area was significantly attenuated in transplanted groups compared to those in control. (Control 2.44 ± 0.07, Transplant 1.85 ± 0.13, *p* = 0.002, one-way ANOVA with Bonferroni test, Figure 5C), and there were no significant differences in GFAP immuno-intensity among the transplant groups (Figure 5C). Analysis of the fibrotic scar volume determined by PDGFRβ^+^ area revealed that fibrotic scar size in the transplanted lesion was significantly smaller than the control (Figure 5F, Control 0.169 mm^3^, NPC 0.054 mm^3^, *p* = 0.012, *t*-test). These data indicated that transplants of dNPCs resulted in reduced glial/fibrotic scar by replacing lost host cells, and dNPCs modified glial/fibrotic scar characteristics by changing scar composition.

### 3.4. NPC Transplants Promote Host Sensory Axon Regeneration

Our previous studies have demonstrated that transplantation of NPCs promoted sensory axon regeneration and provided a permissive environment for axonal growth. To address the ability of dNPCs to promote axon regeneration in chronic injury, we traced ascending sensory axons with CTB by injecting into sciatic nerve three days before sacrifice. Most CTB^+^ axons in control groups were found caudal to the lesion border (Figure 6A) suggesting axon dieback, with some ending within 1 mm of the caudal lesion border. In contrast, animals with dNPCs transplant 4 weeks after injury showed graft extending from the lesion border to greater than several hundred micrometers into the graft, and almost all CTB^+^ axons extended within the transplant area (Figure 6B). We analyzed the sensory axon regeneration (percentage of CTB^+^ axons at different locations and longest axon growth into the lesion site) in two ways: (1) 4 weeks delay for the transplant and survival for 3 and 5 weeks, respectively, and (2) transplant at different delay time and survival for 5 weeks. To examine whether sensory axon regeneration was affected by survival time after transplantation, a 4-week delay for transplant was selected and axon growth was analyzed at 3 and 5 weeks after transplantation. Higher percentages of CTB^+^ axons were seen inside the lesion/transplant area as well as near the caudal lesion border in transplant groups compared to the control group. However, survival time did not change axon growth in either control or transplant groups (Figure 6C), except at 0.2 mm inside the lesion/transplant, more CTB^+^ axons were present at 5 weeks than at 3 weeks after transplant. Quantitative analysis showed that 62.1 ± 10.5% of total CTB^+^ axons penetrated across the caudal lesion border into the lesion/transplant area in transplanted animals, while only 5.4 ± 1.1% in control animals grew into the lesion from the caudal lesion border. (*p* < 0.001, Two Way ANOVA with Holm-Sidak test, Figure 6C). Sensory axons grew a longer distance inside the lesion/transplant region in both 3- and 5-week survival time than the control groups but transplant survival time did not change the distance that axons grew inside the lesion/transplant (Figure 6D). To examine whether sensory axon regeneration was influenced by different delay times for transplant, 5 weeks survival after cell transplant was selected and axon growth was analyzed at 4-, 6-, and 12-week delay for transplantation. The distance sensory axons grew into the lesion/transplant area was not significantly different compared among all transplant groups by the different delay times for transplantation (Figure 6D). Regenerating axons extended as long as 0.5 mm into the lesion center, but none of the CTB^+^ axons extended across rostral lesion border (Figure 6D).

## 4. Discussion

Our previous studies demonstrated that when NPCs are transplanted into an acute dorsal column lesion site, cells survived within the lesion site and differentiated into both neurons and astrocytes. Host sensory axons grew into the transplants and made synapses with neurons derived from transplants. These neurons also sent their axons out of the transplants and followed the pathway created by BDNF-expressing lentivirus to the target dorsal column nuclei (DCN), which reconnected the sensory pathway by forming a functional relay [26]. Here we tested whether this method can be used in a chronic SCI model and focused on analysis of cell survival and axonal growth after transplanting NPC into a chronic dorsal column lesion at different delayed time points with different survival times. We found that transplanted dNPCs (dissociated) but not cNPCs (cultured) survived at lesion area in all delayed transplant time points. Transplanted dNPCs differentiated into mature neurons and astrocytes, extended axons rostrally and caudally, and attracted sensory axons growing into the transplants with no significant differences among the various delayed times.

Survival of transplanted cells is critical for the potential efficacy of cell transplants in SCI and may be dependent on the location of cell delivery, the type of cells employed and the objectives of the study. Numerous cell transplant studies injected cells into the spinal cord rostral and caudal to the injury, but not into the injury site to avoid the toxic environment [43,44,45]. We reasoned that having the transplant within the injury will reduce the inhibitory nature of the injury environment and modulate the secondary injury. We have demonstrated this concept in various injury models including dorsal column lesion [26,38], hemisection [37], transection [35], and contusion [42,46]. The timing of transplantation is also an important variable, and most studies use a subacute delay of about a week, again to avoid the acute stages of the injury associated with cell death and oxidative damage. We reasoned that acute transplantation avoids a second invasive damage and will test cell survival and integration under the most severe conditions at the time of injury. Accordingly, we have shown cell survival of cNPCs in acute dorsal column lesion and lateral funiculus lesion, as well as in subacute contusion. However, in a transection model, few cNPCs survived at the lesion site in either acute or subacute stages, while dNPCs showed good cell survival even when transplanted into the acute injury site [35]. When cNPCs were transplanted into the injury area 13 weeks after a contusion injury, the survival of transplanted cells was highly variable ranging from transplants present in the lesion area to no survival at all [46]. In this study, we found that dNPCs showed much better survival than cNPCs within the dorsal column lesion area when transplanted 4–12 weeks after SCI with no significant differences amount the groups. Better survival of dNPCs compared to cNPCs may be due to the properties of the cell preparation. For example, dNPCs were prepared from E13.5 AP rat spinal cord by mechanically dissociation, without digestion of extracellular components, providing a rich and physiologically supportive environment for cell survival and differentiation. The dNPCs also included other non-neural cells providing a composition that is similar to the embryonic spinal cord. Importantly, acute transplants of dNPCs in the chronic dorsal column lesion (up to 12 weeks) survived well in the lesion site without any addition of growth factors or matrix molecules, which is different from other studies using NPCs with a complex growth factors cocktails and matrix [47,48].

NPCs derived from E13.5 spinal cords contain neuronal and glial restricted progenitor (NRPs and GRPs, respectively) and when transplanted into the intact or acutely injured spinal cord differentiate into neurons and glia, mostly astrocytes [37,38,46]. In agreement with these findings, we now found that transplantation dNPCs into a chronic dorsal column lesion area resulted in differentiation of cells into mature neurons, astrocytes, and oligodendrocytes. About half of them became astrocytes, 20% neurons or oligodendrocytes, while the rest remained undifferentiated. Neurons derived from the transplants expressed both excitatory and inhibitory phenotypes. The presence of mature neurons and young astrocytes in the chronic injury, suggests the potential for formation of a functional relay and restored connectivity. Indeed, in support of connectivity we found that axons from transplant-derived neurons grew out of the transplant in both rostral and caudal directions indicating that these neurons are intrinsically insensitive to chondroitin sulfate proteoglycans (CSPGs) due to low levels of receptor expression supported by permissive factors secreted by the astrocytes, which together reduce the inhibitory effects of CSPGs and promote axonal growth [49].

The environment of chronic injury is complex with a dense glial scar around the lesion area and a cavity filled with fibroblasts, macrophages, and microglia [50,51]. This environment provides a challenge to transplant survival inside the lesion area and to axon growth into and out of the injury site. Most cell transplantations were injected into rostral and caudal to the injury site so that transplanted cells could survive better, but without transplanted cells inside the lesion, there is no replacement of lost cells inside the lesion, thus no bridge that could attract/guide the damaged host axons to grow into/through. We found that when the dNPCs were transplanted into the chronic dorsal column lesion site, the cells survived and significantly reduced the glial scar around the injury site as well as the fibrotic scar within the injury site. We suggest that NPC-derived GRPs can play a major role on reducing the glial scar based on previous studies in which enriched GRPs were transplanted into acute or sub-acute contusion lesion modified the lesion environment and reduced astrocytic scarring [52,53,54]. Taken together it appears that either NPCs or GRPs alone, when transplanted into the injured spinal cord at any stages of SCI, can modify the lesion environment and reduce the glial scar.

Damaged axons do not regenerate after injury in the adult mammalian CNS. However, they can regenerate into the lesion area if the environment of injury area; is modified by growth factors [55], free of inhibitory molecules [7], receives a transplant of cells [18], or biomaterials [56] at or around lesion area. At the same time, it is also important to enhance the intrinsic capability of neurons to regenerate [10,57,58,59] including corticospinal, brainstem and sensory tracts. We focused on sensory axon regeneration following our previous studies which demonstrated that sensory axons labeled by CTB can grow into transplants which were acutely transplanted into a dorsal column lesion [26,33,38]. At 4 weeks delay we found that transplants of dNPCs significantly increased the number and the distance of axon growth into the lesion area relative to control injury, with over 500 µm of the longest axon growth, which represents the middle of the injury/transplant area. This growth was present both rostrally and caudally, remained stable throughout the 5 weeks of analysis and in transplantation delays of delays of 4–12 weeks. These results indicate that transplanted dNPCs at various chronic times following SCI shows robust efficacy with respect to cell survival, differentiation, and the capacity for axon growth.

Other studies have tried various interventions to promote sensory axon regeneration after SCI. Transplanted autologous bone marrow stromal cells with NT-3 injected within and rostral to the graft combined with cAMP injected into L4 dorsal root ganglion promoted sensory axons growing out of the lesion/transplant [60]. When using antibodies against NG2 proteoglycan, sensory axons regenerated into dorsal column lesion. Furthermore, when NG2 antibody is combined with a peripheral nerve conditioning lesion, sensory axons grew past the glial scar and into the white matter rostral to the injury site [61]. A recent study showed that continuous delivery of NT-3 using a tet-off lentiviral gene construct injected rostral to the graft of bone marrow stromal cells promote CTB labeled sensory axon growth into and beyond the lesion/graft site. Interestingly, axonal regeneration declined when NT-3 delivery was turned off [62]. All these studies demonstrate that combined strategies can promote sensory axon regeneration. However, none of these studies showed re-connection of the sensory pathway. Our previous study in acute SCI indicates that instead of promoting long distance axonal regeneration, damaged sensory pathways can be reconstructed by a combination strategy that forms a functional relay. To form such a relay, several specific steps are needed to make sure that: (1) the graft survives and generates neurons, (2) axon growth occurs into and out of the graft by host axons and neurons derived from the graft, respectively, (3) there is formation of physiologically active synaptic connections and restoration of function [31,63]. In this study, we have demonstrated several elements for relay formation in the chronic stage: survival of transplanted dNPCs and neuronal differentiation, axonal growth into and out of the graft by host sensory axons and neurons derived from the graft. Our next steps will investigate the formation of functional synapses and guidance of axons growing toward their targets.

## 5. Conclusions

We have demonstrated that dNPCs transplanted into a chronic dorsal column lesion can survive well at different delayed transplant times up to 12 weeks after injury. Transplanted cells differentiate into mature neurons and astrocytes as well as oligodendrocytes. Axonal growth into and out of the transplant is seen from host sensory axons and neurons derived from the transplant. All these data provide basic components for relay formation in the chronic stage. Our next experiments will use the dorsal column system to investigate (1) whether the first synapse forms between host sensory axons and transplant derived neurons. (2) whether the second synapse forms between grafted neurons and neurons in the dorsal column nuclei and if they can be guided by growth factor support to form a functional relay.

## Figures and Tables

**Figure 1 biomedicines-10-00350-f001:**
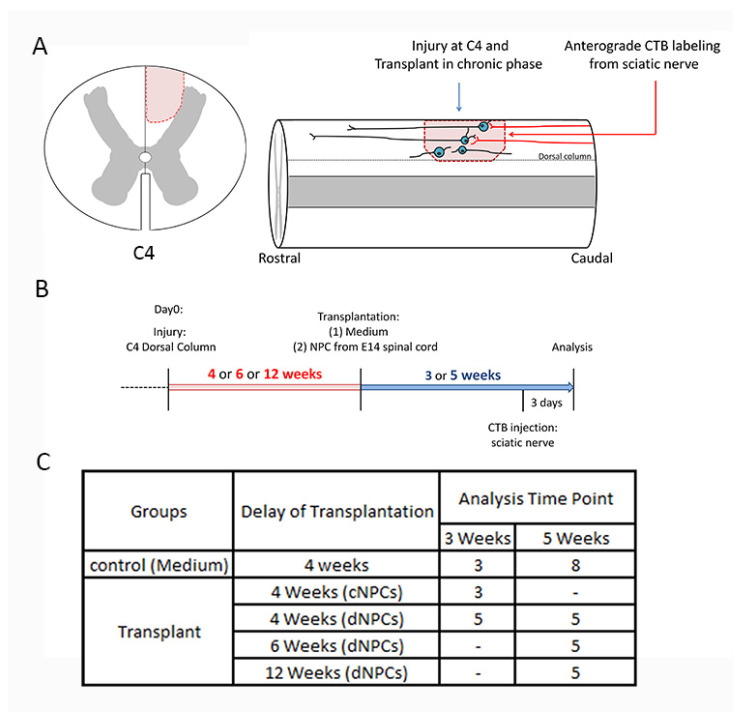
Outline of experimental design, timeline, and groups. The left panel (**A**) shows the location of the dorsal column lesion at C4 cross spinal cord section. The right panel (**A**) shows a cartoon image of experiments. (**B**) demonstrates the experimental design. Rats received a unilateral C4 dorsal column injury an appropriate period of time before cell transplantation. Neural progenitor cells (NPCs), either cNPCs or dNPCs were transplanted within the lesion site at 4, 6, 12 weeks (red highlight) after the initial injury. Tracing sensory axon growth with CTB was conducted 3 days before the end of experiment. Animals survived for 3 or 5 weeks (blue highlight) after transplantation. (**C**) shows experimental groups. cNPCs: cultured NPCs from stock of E13.5 AP transgenic rat spinal cord. dNPCs: dissociated spinal cord of E13.5 AP transgenic rat. Preparation of both cNPCs and dNPCs did not include any added growth factors.

**Figure 2 biomedicines-10-00350-f002:**
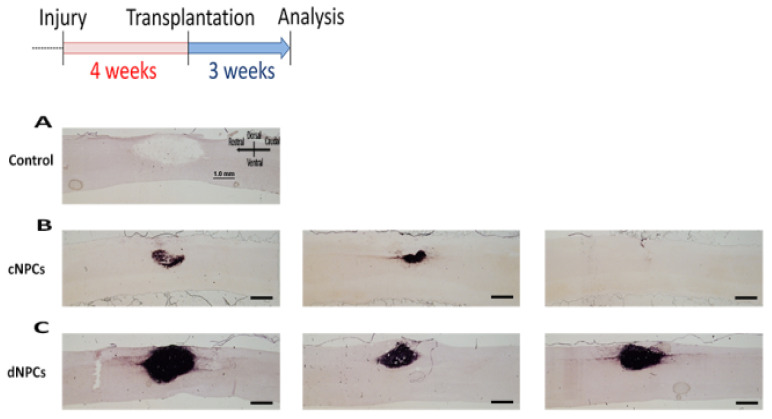
NPC transplants survival. Two types of cells were injected into the lesion site 4 weeks after injury and survived for 3 weeks after transplantation. AP histochemical staining demonstrated that there was no AP^+^ staining in the control group (**A**). In cNPCs group, transplanted cells did not show good survival with some AP^+^ cells inside the lesion but not filling the lesion area and some animals with no AP^+^ cells inside the lesion/transplant area (**B**). In dNPCs group, transplanted cells survived in all rats and filled the lesion/transplant area (**C**). Scale bar = 1.0 mm.

**Figure 3 biomedicines-10-00350-f003:**
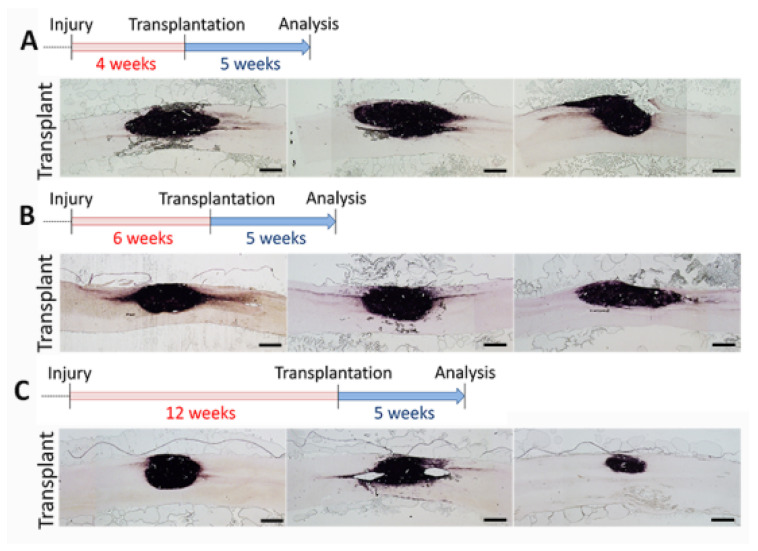
dNPC transplant survival at different delay time points after SCI. AP histochemical staining demonstrates good cell survival 5 weeks after cell transplant in different injury stages: 4 weeks (**A**), 6 weeks (**B**), and 12 weeks, (**C**) after SCI. Three images from each time point represent 3 individual rats from each group. AP^+^ axons from the transplant extend out of the transplant in both rostral and caudal directions in almost all rats. Scale bar = 1.0 mm.

**Figure 4 biomedicines-10-00350-f004:**
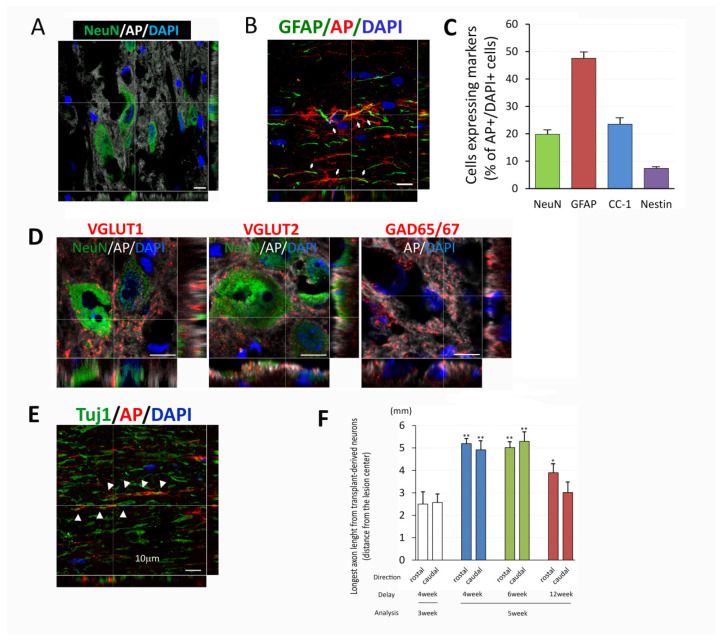
Cell differentiation and axon growth from NPC transplants. At different delay time points and survival for either 3 or 5 weeks transplanted cells inside the chronic injury area differentiated into mature neurons (**A**), astrocytes (**B**), oligodendrocytes, and some nestin^+^ undifferentiated NPCs (Images not shown). Measurement of percentage of differentiated cells inside the transplant indicates that about 50% of transplanted cells differentiated into astrocytes (GFAP^+^), 20% into neurons (NeuN^+^) and oligodendrocytes (CC-1^+^), and less than 10% of undifferentiated NPCs (Nestin^+^) (**C**). Mature neurons derived from transplants also expressed VGLUT 1^+^ and 2^+^ (excitatory) and GAD65/67^+^ (inhibitory) neuronal markers (**D**). Axons derived from transplants extended long distances from center of the transplants in both rostral and caudal directions. Double staining of AP and Tuj1 indicated that many of AP^+^ fibers were co-localized with Tuj1 (**E**). Analysis of extension of AP^+^ fibers (**F**) demonstrated longer extension in both directions in 3 delay groups for 5 weeks after transplant compared to 3 weeks survival in 4 weeks delay. Data are presented as mean ± SEM. Statistical significance was set at *p* < 0.05 for all comparisons (One-way ANOVA with Bonferroni test). Asterisk indicates significant difference comparing to group at 3 weeks post transplantation with a delay of 4 weeks (* *p* < 0.05, ** *p* < 0.005). Scale bar = 10 µm (**A**,**B**,**D**,**E**).

**Figure 5 biomedicines-10-00350-f005:**
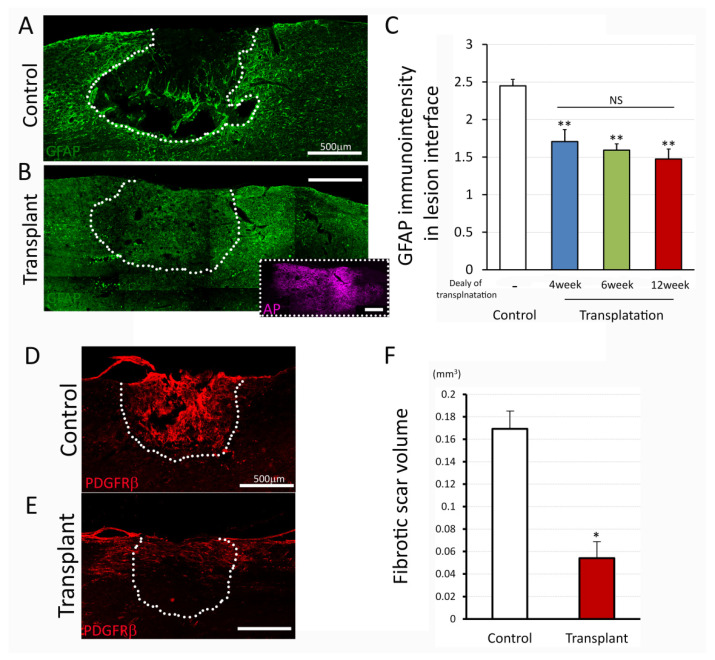
Glial and fibrotic scar after injury and dNPCs transplantation. (**A**–**C**) demonstrate GFAP immuno-intensity. Immunostaining represents the apparent GFAP-reactive boundary in control (**A**) and attenuation of immuno-intensity in the transplanted lesion interface at 5 weeks post transplantation (**B**). Quantitative analysis (**C**) shows significant attenuation of the GFAP immuno-intensity in three delay transplant groups compared to control group. Perivascular fibroblasts marker, PDGFRβ was used to estimate fibrotic scar volume (**D**–**F**). Immunostaining with PDGFRβ shows the fibrotic scar filled by the predominant PDGFRβ^+^ cells in the lesion site in control (**D**) and reduced immunoreactivity in transplant group (**E**) at 5 weeks post transplantation with delay of 12 weeks. Estimated fibrotic scar volume is shown in panel (**F**) with significant reduction of fibrotic scar in transplant group compared to control group. Data are presented as mean ± SEM. Statistical significance was set at *p* < 0.05 (**C**; one-way ANOVA with Bonferroni test, **F**; *t*-test). Asterisk indicates significant difference comparing to control (* *p* < 0.05, ** *p* < 0.005). NS: not significant. Scale bar = 500 µm. Lesion/transplant area is identified with a white dish-line.

**Figure 6 biomedicines-10-00350-f006:**
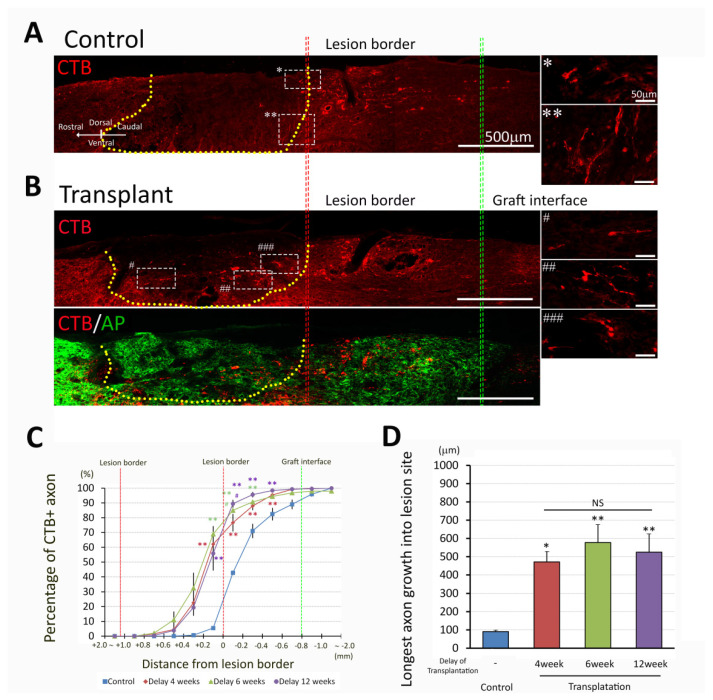
Sensory axon regeneration after NPC transplantation. Most axons in control group (**A**) stopped (*) or changed direction (**) at the lesion boundary (insert with higher magnification), while CTB-labeled axons extended into the lesion site in transplanted group (**B**) at 5 weeks post transplantation with delay of 4 weeks (Signs #, ##, ### indicate the locations of white dish-line boxes in the left panel B which match the insert with a higher magnification in the right panel B). Double labeling of the CTB and AP markers indicates that there are CTB^+^ axons throughout the graft (**B**). Quantitative analysis of the distribution of CTB^+^ axons (**C**) and longest axon growth into the lesion/transplant site (**D**) were compared with 4 weeks delayed transplant and survival for 3- and 5-week post transplantation. Both transplanted groups showed a higher percentage of CTB^+^ axons 200 µm caudal to the lesion border as well as 200 µm and 400 µm within the lesion area (**C**, **, *p* < 0.005 compared to their respective controls). Furthermore at 200 µm within the lesion area, the 5 weeks delayed transplantation group demonstrated a significantly greater proportion of CTB^+^ axons than the 3 weeks delayed transplantation group (#, *p* < 0.05). Data are presented as mean ± SEM. Statistical significance was set at *p* < 0.05 (**C**; two-way ANOVA with Bonferroni post-hoc test, **D**; one-way ANOVA with post-hoc Bonferroni test). * *p* < 0.05; ** *p* < 0.005. NS: no significant difference compared to transplantation groups at 4, 6, and 12 weeks delayed after lesion. Lesion/transplant area is identified with a yellow dish-line.

**Table 1 biomedicines-10-00350-t001:** Primary and secondary antibodies used in this study.

Name	Type	Dilution	Source
AP	Mouse IgG1	1:400	Chemicon (Temecula, CA, USA)
AP	Rabbit IgG	1:1000	Serotec (Hercules, CA, USA)
NeuN	Mouse IgG1	1:100	Chemicon
GFAP	Mouse IgG1	1:1000	Chemicon
GFAP	Rabbit	1:2000	Chemicon
CC-1	Mouse IgG	1:100	Chemicon
Nestin	Mouse IgG1	1:1000	BD Pharmigen (San Diego, CA, USA)
VGLUT1	Guinea Pig IgG	1:10,000	Chemicon
VGLUT2	Guinea Pig IgG	1:2500	Chemicon
GAD65/67	Rabbit IgG	1:500	Chemicon
Tuj1	Rabbit IgG	1:1000	Covance (Princeton, NJ, USA)
PDGFR	Rabbit IgG	1:200	Abcam (Waltham, MA, USA)
Choleragenoid (CTB)	Goat	1:2000	List Biological Laboratories (Campbell, CA, USA)
Synaptophysine	Guinea Pig IgG	1:500	Sy synaptic system (Goettingen, Germany)
goat anti mouse IgG Alexa Fluor 594		1:400	Life technologies (Carlsbad, CA, USA)
goat anti rabbit Rhodamin Red		1:400	Jackson (West Grove, PA, USA)
goat anti mouse-IgG FITC		1:400	Jackson
goat anti rabbit FITC		1:400	Jackson
goat anti mouse-IgG Alexa Fluor 647		1:400	Life technologies
donkey anti goat-IgG Rhodamin Red		1:400	Jackson
donkey anti guinea pig-IgG Rhodamin Red		1:400	Jackson
donkey anti mouse-IgG Alexa Fluor 488		1:400	Life technologies
donkey anti rabbit-IgG Alexa Fluor 488		1:400	Jackson
donkey anti mouse-IgG Alexa Fluor 647		1:400	Life technologies

## Data Availability

The data in this study are available on request from the corresponding authors.

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
