# Peer review of "Transplanting Neural Progenitor Cells into a Chronic Dorsal Column Lesion Model"

_biomedicines, 2022, doi:10.3390/biomedicines10020350_

Round 1

Reviewer 1 Report

In this paper, the authors reported the NPC transplantation on chronic SCI rodent model and showed the regeneration of neuron and axon. This is a very interesting study with clinical implication. From the perspective of academic criticism, several technical concerns need to be addressed to further improve the quality of this manuscript, as appended below.

A section discussing the state-of-art of regenerative medicine for spinal cord injury should be included in the introduction.

What is the rationale of transplanting 1 million NPC per rat? Was different dose of transplant tested? In Figure 5 and 6, the amount of GFAP expression and the extension of axons did not significantly increase after 4 weeks. Would this be correlated to the limited transplanted cells? Please discuss this in the results or discussion.

Based on the method, the experiments were carried out with only one line of cNPCs and one line of dNPCs, which is not ideal for eliminating the individual effects. The lack of biological repeats would be a key threat to the significance of the presented results. Please address this issue by providing extra data or proper explanation.

As the author compared cNPCs and dNPCs with their performance in in vivo study, having some basic characterization of the cells, for example, immunoassay or qPCR of stem cell markers, would further support the argument.

For all the IHC images of transplant, it would be helpful for the visualization if the contour of the lesion edge was drawn. Alternative, a quantitative measurement of the lesion area with statistical analysis showing the regeneration over time would make the data much more convincing.

Scale bars should be included in all the IHC/IF images.

Figure 5, 6 should be replaced by figures with higher resolution.

Author Response

Reviewer1:

In this paper, the authors reported the NPC transplantation on chronic SCI rodent model and showed the regeneration of neuron and axon. This is a very interesting study with clinical implication. From the perspective of academic criticism, several technical concerns need to be addressed to further improve the quality of this manuscript, as appended below.

 As suggested, we added a general discussion on regenerative medicine in Introduction.

Recent progress in cell and molecular biology identified mechanisms associate with the regenerative program in adult CNS neurons as well as the signaling associated with the inhibitory effects of Chondroitin Sulfate Proteoglycan and myelin paving a way for improved strategies to promote regeneration and repair after injury (He and Jin, 2016, Curcio and Bradke 2018, Tedeschi and Popovich, 2019, Fawcett, 2020).

What is the rationale of transplanting 1 million NPC per rat? Was different dose of transplant tested? In Figure 5 and 6, the amount of GFAP expression and the extension of axons did not significantly increase after 4 weeks. Would this be correlated to the limited transplanted cells? Please discuss this in the results or discussion.

The rationale of transplanting 1 million NPC per rat is based on our previous studies in which we optimized the number of cells for survival, integration and connectivity (see papers below). To clarify: what we found is that the longest axons from graft derived neurons increased between 3 and 5 weeks (Fig. 4). However, the longest axons growing into the transplant did not change between 3 and 5 weeks. Our interpretation of these results was that while the growth capacity of NPC neurons derived from embryonic spinal cord is high, the regenerative capacity of host neurons is limited and may need additional strategies to promote robust regeneration. Furthermore, there were no significant changes of GFAP expression after 4 weeks indicating that the transplanted cells and lesion area are correlated with cell survival in lesion area.

Bonner JF, Blesch A, Neuhuber B, Fischer I. 2009. Promoting directional axon growth from neural progenitors grafted into the injured spinal cord. Journal of Neuroscience Research 88(6):1182-92. PMCID: PMC2844860.

Bonner JF, Connors TM, Silverman WF, Kowalski DP, Lemay MA, and Fischer I. 2011. Grafted neural progenitors integrate and restore synaptic connectivity across the injured spinal cord. J Neuroscience. 31:4675-4686. PMCID: PMC3148661

Medalha CC, Y Jin, T Yamagami, C Haas, I Fischer. 2014. Transplanting neural progenitors into a complete transection model of spinal cord injury. J Neurosci Res. 92(5):607-18..

Based on the method, the experiments were carried out with only one line of cNPCs and one line of dNPCs, which is not ideal for eliminating the individual effects. The lack of biological repeats would be a key threat to the significance of the presented results. Please address this issue by providing extra data or proper explanation.

Neither the cNPCs nor the dNCPs are cell lines, both are primary neuronal cultures. The cNPC, which are culture and keep in vials of frozen stocks are prepared from E13.5 spinal cord in a procedure described in Methods and our previous publications. We prepared and tested dozens of these frozen stocks and in fact used several different vials for our experiments. The dNCP are primary cells that are not cultured at all, and here again we needed several batches to obtain enough cells for the transplantation experiments.

As the author compared cNPCs and dNPCs with their performance in in vivo study, having some basic characterization of the cells, for example, immunoassay or qPCR of stem cell markers, would further support the argument.

As noted above we have prepared and characterized these cells in many of our publications as well as others (e.g., Michael Lane, Mark Tuszynski). Briefly these cells contain both neuronal restricted progenitors (NRPs) and glial restricted progenitors. The NRPs are composed of both excitatory and inhibitory interneurons and importantly, are less sensitive to the inhibitory effects of Chondroitin Sulfate Proteoglycan (See et al, 2010, J Neurotrauma 27(5):951-57, Ketschek et al., 2012, Exp Neurol. 235:627-37).

For all the IHC images of transplant, it would be helpful for the visualization if the contour of the lesion edge was drawn. Alternative, a quantitative measurement of the lesion area with statistical analysis showing the regeneration over time would make the data much more convincing.

We draw a line along the edge of lesion. But we did not include the measurement of the lesion area in this paper since we are focused on the transplanted cells survival and differentiation in different time of chronic injury and the effects on axons growth into and out of the transplants. In future studies we plan to analyze the effects on neuroprotection and restoration of function.

Scale bars should be included in all the IHC/IF images.

We added scale bars in all IHC/IF images that missed.  

Figure 5, 6 should be replaced by figures with higher resolution.

We did as reviewer requested.

Reviewer 2 Report

In this manuscript, the authors explore the efficacy of different neural progenitor cells (NPCs) for chronic spinal cord injury repair. The survival and differentiation of the NPCs, which were transplanted at different times post-injury, were investigated. Overall, this is an interesting work with well-presented results. I suggest acceptance of this manuscript after addressing the following minor issues.

  1. Please describe the surgical methods for the dorsal column lesion and quantify the consistency of lesion size.
  2. Besides the images of transplanted cells in Fig. 2 and Fig. 3, quantification of AP+ area would help for better demonstration.
  3. In Fig. 5, it would be great if the authors can illustrate the edges of lesions.

Author Response

Reviewer2:

In this manuscript, the authors explore the efficacy of different neural progenitor cells (NPCs) for chronic spinal cord injury repair. The survival and differentiation of the NPCs, which were transplanted at different times post-injury, were investigated. Overall, this is an interesting work with well-presented results. I suggest acceptance of this manuscript after addressing the following minor issues.

  1. Please describe the surgical methods for the dorsal column lesion and quantify the consistency of lesion size.

We added dorsal column lesion in details in the method.

  1. Besides the images of transplanted cells in Fig. 2 and Fig. 3, quantification of AP+ area would help for better demonstration.

We added quantification of AP+ area Fig 2.  Fig.3 has shown quantification of AP+ area in the manuscript.

  1. In Fig. 5, it would be great if the authors can illustrate the edges of lesions.

We added line along the edges of lesions in Fig 5.

Round 2

Reviewer 1 Report

The reviewer thank the authors for the reply.

Re-"Neither the cNPCs nor the dNCPs are cell lines, both are primary neuronal cultures. The cNPC, which are culture and keep in vials of frozen stocks are prepared from E13.5 spinal cord in a procedure described in Methods and our previous publications. We prepared and tested dozens of these frozen stocks and in fact used several different vials for our experiments. The dNCP are primary cells that are not cultured at all, and here again we needed several batches to obtain enough cells for the transplantation experiments."

The reviewer understands the fact that all the NPCs used were primary cells, which was exactly why the individual effects would be an issue here. According to the author's reply, it seems like the NPCs from different donors were pooled together and used for the transplantation.If so, the number of NPC vials used in each pool, the criteria for cell selection, and a description of the process should be added to the Method. Another important question here is, are the data collected from multiple mice received identical cell transplantation from one cell pool or multiple cell pools? If all the animal data were generated from one NPC pool for each group, the results could be significantly biased by the individual effects.

Re-"As noted above we have prepared and characterized these cells in many of our publications as well as others (e.g., Michael Lane, Mark Tuszynski). Briefly these cells contain both neuronal restricted progenitors (NRPs) and glial restricted progenitors. The NRPs are composed of both excitatory and inhibitory interneurons and importantly, are less sensitive to the inhibitory effects of Chondroitin Sulfate Proteoglycan (See et al, 2010, J Neurotrauma 27(5):951-57, Ketschek et al., 2012, Exp Neurol. 235:627-37)."

The reviewer was questioning if the difference found in animal study was actually caused by the difference in NPCs since there was no direct characterization on the cells. If the experiments were not performed with the exact same vials of cell reported in the mentioned reference, the characterization data could not be directly referred to the current work.  It would not be a rigorous study if there is no basic characterization data provided for the cells used in transplantation.

Author Response

Response to reviewer 1 

Re-"Neither the cNPCs nor the dNCPs are cell lines, both are primary neuronal cultures. The cNPC, which are culture and keep in vials of frozen stocks are prepared from E13.5 spinal cord in a procedure described in Methods and our previous publications. We prepared and tested dozens of these frozen stocks and in fact used several different vials for our experiments. The dNCP are primary cells that are not cultured at all, and here again we needed several batches to obtain enough cells for the transplantation experiments."

The reviewer understands the fact that all the NPCs used were primary cells, which was exactly why the individual effects would be an issue here. According to the author's reply, it seems like the NPCs from different donors were pooled together and used for the transplantation.If so, the number of NPC vials used in each pool, the criteria for cell selection, and a description of the process should be added to the Method. Another important question here is, are the data collected from multiple mice received identical cell transplantation from one cell pool or multiple cell pools? If all the animal data were generated from one NPC pool for each group, the results could be significantly biased by the individual effects.

Re-"As noted above we have prepared and characterized these cells in many of our publications as well as others (e.g., Michael Lane, Mark Tuszynski). Briefly these cells contain both neuronal restricted progenitors (NRPs) and glial restricted progenitors. The NRPs are composed of both excitatory and inhibitory interneurons and importantly, are less sensitive to the inhibitory effects of Chondroitin Sulfate Proteoglycan (See et al, 2010, J Neurotrauma 27(5):951-57, Ketschek et al., 2012, Exp Neurol. 235:627-37)."

The reviewer was questioning if the difference found in animal study was actually caused by the difference in NPCs since there was no direct characterization on the cells. If the experiments were not performed with the exact same vials of cell reported in the mentioned reference, the characterization data could not be directly referred to the current work.  It would not be a rigorous study if there is no basic characterization data provided for the cells used in transplantation.

We appreciate the comments of the reviewers on the importance of clarity on the sources of NPCs, how they are collected, pooled and transplanted. Consequently, we added more details to the Methods section relative to the two different types of NPCs.

The cNPC were prepared from multiple embryos (E13.5 spinal cords, containing the stable alkaline phosphatase reporter gene (AP+) ) and pooled following short culturing and quality control for the composition of neuronal and glial precursors (NRPs and GRPs) at a ratio of 40:60 with >5% of either mature neural cells or non-neural cells.  They are then kept in vials of frozen stocks containing 3-5 million cells. We have a very detailed description of the process and characterization of the cells in a book chapter we added to the reference list.

Bonner JF, C Haas, and I Fischer. “Preparation of Neural Stem Cells and Progenitors: Neuronal Production and Grafting Applications.” (2013). Methods in Molecular Biology – Neuronal Cell Culture. Volume 1078. Editors: Amini, Shohreh; White Martyn L. Methods Mol Biol. 1078:56-88.

As for dNCP, which as the reviewer noted are primary cultures, here again we have to use multiple pooled embryos (E13.5 spinal cords) for each transplantation. As these cells are not cultured or processed, they are in fact fetal tissue and represent the exact composition of the in vivo tissue. We described the preparation method and in vivo characterization in several papers with reference to the original methods pioneered by Reier and colleagues.

Lepore A, Fischer I. (2005). Lineage-restricted neural precursors survive, migrate and differentiate following transplantation into the injured adult spinal cord. Exp Neurol 194:230-242.

Medalha CC, Y Jin, T Yamagami, C Haas, I Fischer. 2014. Transplanting neural progenitors into a complete transection model of spinal cord injury. J Neurosci Res. 92(5):607-18

Round 3

Reviewer 1 Report

Thanks for the update on the cell collection methods. But the problem of individual effects and the lack of in vitro characterization were not addressed in the response.